# Auditing Differentially Private Machine Learning: How Private is Private SGD?

**Matthew Jagielski**
Northeastern University
jagielski@ccs.neu.edu

**Jonathan Ullman**
Northeastern University
jullman@ccs.neu.edu

**Alina Oprea**
Northeastern University
a.oprea@northeastern.edu

## Abstract

We investigate whether Differentially Private SGD offers better privacy in practice than what is guaranteed by its state-of-the-art analysis. We do so via novel data poisoning attacks, which we show correspond to realistic privacy attacks. While previous work (Ma et al., arXiv 2019) proposed this connection between differential privacy and data poisoning as a defense against data poisoning, our use as a tool for understanding the privacy of a specific mechanism is new. More generally, our work takes a quantitative, empirical approach to understanding the privacy afforded by specific implementations of differentially private algorithms that we believe has the potential to complement and influence analytical work on differential privacy.

## 1 Introduction

Differential privacy [DMNS06] has become the de facto standard for guaranteeing privacy in machine learning and statistical analysis, and is now being deployed by many organizations including Apple [TVV+17], Google [EPK14, BEM+17, PSM+18], and the US Census Bureau [HMA+17]. Now that differential privacy has moved from theory to practice, there has been considerable attention on optimizing and evaluating differentially private machine learning algorithms, notably differentially private stochastic gradient descent (henceforth, DP-SGD) [SCS13, BST14, ACG+16], which is now widely available in TensorFlow Privacy [Goo]. DP-SGD is the building block for training many widely used private classification models, including feed-forward and convolutional neural networks.

Differential privacy gives a strong *worst-case* guarantee of individual privacy: a differentially private algorithm ensures that, for any set of training examples, no attacker, no matter how powerful, can learn much more information about a single training example than they could have learned had that example been excluded from the training data. The amount of information is quantified by a *privacy parameter* $\varepsilon$.[1] Intuitively, a smaller $\varepsilon$ means stronger privacy protections, but leads to lower accuracy. As such there is often pressure to set this parameter as large as one feels still gives a reasonable privacy guarantee, and relatively large parameters such as $\varepsilon = 2$ are not uncommon. However, this guarantee is not entirely satisfying, as such an algorithm might allow an attacker to guess a random bit of information about each training example with approximately 86% accuracy. As such there is often a gap between the strong formal protections promised by differential privacy and the specific quantitative implications of the choice of $\varepsilon$ in practice.

This state-of-affairs is often justified by the fact that our analysis of the algorithm is often pessimistic. First of all, $\varepsilon$ is a parameter that has to be determined by careful analysis, and often existing theoretical analysis is not tight. Indeed a big part of making differentially private machine learning practical has been the significant body of work giving progressively more refined privacy analyses specifically

for DP-SGD [ACG+16, DRS19, MTZ19, YLP+19], and for all we know these bounds on $\varepsilon$ will continue to shrink. Indeed, it is provably intractable to determine the tightest bound on $\varepsilon$ for a given algorithm [GM18]. Second, differential privacy is a worst-case notion, as the mechanism might have stronger privacy guarantees on realistic datasets and realistic attackers. Although it is plausible that differentially private algorithms with large values of $\varepsilon$ provide strong privacy in practice, it is far from certain, which makes it difficult to understand the appropriate value of $\varepsilon$ for practical deployments.

## 1.1 Our Contributions

**Auditing DP-SGD.** In this paper we investigate the extent to which DP-SGD,[2] does or does not give better privacy in practice than what its current theoretical analysis suggests. We do so using novel *data poisoning attacks*. Specifically, our method starts with a dataset $D$ of interest (e.g. Fashion-MNIST) and some algorithm $\mathcal{A}$ (e.g. DP-SGD with a specific setting of hyperparameters), and produces a small poisoning set $S$ of $k$ points and a binary classifier $T$ such that $T$ distinguishes the distribution $\mathcal{A}(D)$ from $\mathcal{A}(D \cup S)$ with significant advantage over random guessing. If $\mathcal{A}$ were $\varepsilon$-DP, then $T$ could have accuracy at most $\exp(\varepsilon k)/(1 + \exp(\varepsilon k))$, so if we can estimate the accuracy of $T$ we can infer a lower bound on $\varepsilon$. While previous work [MZH19] proposed to use this connection between differential privacy and data poisoning as a defense against data poisoning, our use in this context of auditing the privacy of DP-SGD is new.

Specifically, for certain natural choices of hyperparameters in DP-SGD, and standard benchmark datasets (see Figure 2), our attacks give lower bounds on $\varepsilon$ that are approximately 10x better than what we could obtain from previous methods, and are within approximately 10x of the worst-case, analytically derived upper bound. For context, previous theoretical improvements to the analysis have improved the worst-case upper bounds by factors of more than 1000x over the naïve analysis, and thus our results show that we cannot hope for similarly dramatic gains in the future.

**Novel Data Poisoning Attacks.** We find that existing data poisoning attacks, as well as membership inference attacks proposed by prior work, have poor or trivial performance not only against DP-SGD, but even against SGD with gradient clipping (i.e. rescaling gradients to have norm no larger than some $C$). Gradient clipping is an important part of DP-SGD, but does not provide any formal privacy guarantees on its own. Thus, we develop a novel data poisoning attack that is more robust to gradient clipping, and also performs much better against DP-SGD.

Intuitively, data poisoning attacks introduce new points whose gradients will change the model in a certain direction, and the attack impact increases when adding poisoning points of larger gradients. Existing attacks modify the model in a random direction, and have to push far enough that the original distribution on model parameters and the new distribution become distinguishable. To be effective, these attacks use points which induce large gradients, making the attack sensitive to gradient clipping. On the other hand, our attack improves by finding the direction where the model parameters have the lowest variance, and select poisoning points that modify the model in that direction. Therefore, we achieve the same effect of model poisoning with poisoning points of smaller gradients, thereby making the attack more robust to clipping.

**The Role of Auditing in DP.** More generally, our work takes a quantitative, empirical approach to *auditing* the privacy afforded by specific implementations of differentially private algorithms. Our auditing algorithm described in Section 2.2 is generic, and can be used with an appropriate poisoning algorithm, like the one we describe for DP-SGD. We do not advocate trying to definitively measure privacy of an algorithm empirically, since it's hopeless to try to anticipate all future attacks. Rather, we believe this empirical approach has the potential to complement and influence analytical work on differential privacy, somewhat analogous to the way *cryptanalysis* informs the design and deployment of *cryptography*.

Specifically, we believe this approach can complement the theory in several ways: (1) Most directly, by advancing the state-of-art in privacy attacks, we can either demonstrate that a given algorithm with a given choice of parameters is not sufficiently private, or give some confidence that it might be sufficiently private. (2) Establishing strong lower bounds on $\varepsilon$ gives a sense of how much more one could hope to get out of tightening the existing privacy analysis. (3) Observing how the performance

of the attack depends on different datasets, hyperparameters, and variants of the algorithm can identify promising new phenomena to explore theoretically. (4) Producing concrete privacy violations can help non-experts interpret the concrete implications of specific choices of the privacy parameter.

## 1.2  Related Work

**DP-SGD.** Differentially private SGD was introduced in [SCS13], and an asymptotically optimal analysis of its privacy properties was given in [BST14]. Notably Abadi et al. [ACG+16] gave greatly improved concrete bounds on its privacy parameter, and showed its practicality for training neural networks, making DP-SGD one of the most promising methods for practical private machine learning. There have been several subsequent efforts to refine the privacy analysis of this specific algorithm [MTZ19, DRS19, YLP+19]. A recent work [HT19] gave a heuristic argument that SGD *itself* (without adding noise to the gradients) satisfies differential privacy, but even then the bounds on $\varepsilon$ are quite large (e.g. $\varepsilon = 13.01$) for most datasets.

**Privacy Attacks.** Although privacy attacks have a very long history, the history of privacy attacks against aggregate statistical information, such as machine learning models, goes back to the seminal work of Dinur and Nissim [DN03] on *reconstruction attacks*. A similar, but easier to implement type of attack, *membership inference attacks*, was first performed by Homer et al. [HSR+08], and theoretically analyzed in [SOJH09, DSS+15]. Shokri et al. [SSSS17] and Yeom et al. [YGFJ18] gave black-box membership inference algorithms for complex machine learning models. Membership inference attacks are compelling because they require relatively weak assumptions, but, as we show, state-of-the-art membership inference attacks lead to quantitatively weak privacy violations. Carlini et al. [CLE+19] show that specific data points, called *canaries*, can be memorized by language models, and that differential privacy with very weak parameters ($\varepsilon > 10^9$) protects from this memorization. Sablayrolles et al. [SDSJ20] show how to modify images to make them "traceable"—to be able to determine whether they were used in a training set.

More directly related to our work, privacy attacks were recently used by Jayaraman and Evans [JE19] to understand the concrete privacy leakage from differentially private machine learning algorithms, specifically DP-SGD. However, the goal of their work is to compare the privacy guarantees offered by different variants of differential privacy, rather than to determine the level of privacy afforded by a given algorithm. As such, their attacks are quantitatively much less powerful than ours (as we show in Figure 2), and are much further from determining the precise privacy guarantees of DP-SGD.

**Differential Privacy and Data Poisoning.** Ma et al. [MZH19] and Hong et al. [HCK+20] evaluate the effectiveness of data poisoning attacks on differentially private machine learning algorithms. Ma et al. consider both the output perturbation and objective perturbation algorithms for learning ridge regression and logistic regression models, proposing attacks on differentially private algorithms, and also argue that differentially privacy can serve as a defense for poisoning attacks. Hong et al. propose differential privacy as a defense for poisoning attacks, showing that DP-SGD performs effectively at defending against existing poisoning attacks in the literature. While differential privacy provides a provable defense for poisoning attacks, our intuition is that the strong poisoning attacks we design allow us to measure a lower bound on the privacy offered by differentially private algorithms.

## 2  (Measuring) Differential Privacy

### 2.1  Differential Privacy Background

We begin by outlining differential privacy and one of its relevant properties: group privacy. We consider machine learning classification tasks, where a dataset consists of many samples from some domain $\mathcal{D} = \mathcal{X} \times \mathcal{Y}$, where $\mathcal{X}$ is the feature domain, and $\mathcal{Y}$ the label domain. We say two datasets $D_0, D_1$ differ on $k$ rows if we can replace at most $k$ elements from $D_0$ to produce $D_1$.

**Definition 2.1** ([DMNS06]). An algorithm $\mathcal{A} : \mathcal{D} \mapsto \mathcal{R}$ is $(\varepsilon, \delta)$-*differentially private* if for any two datasets $D_0, D_1$ which differ on at most one row, and every set of outputs $\mathcal{O} \subseteq \mathcal{R}$:

$$\Pr[\mathcal{A}(D_0) \in \mathcal{O}] \leq e^\varepsilon \Pr[\mathcal{A}(D_1) \in \mathcal{O}] + \delta, \tag{1}$$

where the probabilities are taken only over the randomness of $\mathcal{A}$.

**Lemma 1** (Group Privacy). Let $D_0, D_1$ be two datasets differing on at most $k$ rows, $\mathcal{A}$ is an $(\varepsilon, \delta)$-differentially private algorithm, and $\mathcal{O}$ an arbitrary output set. Then

$$\Pr[\mathcal{A}(D_0) \in \mathcal{O}] \leq e^{k\varepsilon} \Pr[\mathcal{A}(D_1) \in \mathcal{O}] + \frac{e^{k\varepsilon}-1}{e^{\varepsilon}-1} \cdot \delta. \tag{2}$$

Group privacy will give guarantees for poisoning attacks that introduce multiple points.

**DP-SGD.** The most prominent differentially private mechanism for training machine learning models is differentially private stochastic gradient descent (DP-SGD) [SCS13, BST14, ACG$^+$16]. DP-SGD makes two modifications to the standard SGD procedure: clipping gradients to a fixed maximum norm $C$, and adding noise to gradients with standard deviation $\sigma C$, for a given $\sigma$, as shown in Algorithm 1. Given the hyperparameters – clipping norm, noise magnitude, iteration count, and batch size – one can analyze DP-SGD to conclude that it satisfies $(\varepsilon, \delta)$-differential privacy for some parameters $\varepsilon, \delta \geq 0$.

---

**Algorithm 1:** DP-SGD

**Data:** Input: Clipping norm $C$, noise magnitude $\sigma$, iteration count $T$, batch size $b$, dataset $D$, initial model parameters $\theta_0$, learning rate $\eta$

**For** $i \in [T]$
    $G = 0$
    **For** $(x, y) \in$ *batch of b random elements of D*
        $g = \nabla_\theta \ell(\theta_i; (x, y))$
        $G = G + b^{-1}g \cdot \min(1, C||g||_2^{-1})$
    $\theta_i = \theta_{i-1} - \eta(G + \mathcal{N}(0, (C\sigma)^2 \mathbb{I}))$
**Return** $\theta_T$

---

## 2.2 Statistically Measuring Differential Privacy

In this section we describe our main statistical procedure for obtaining lower bounds on the privacy parameter for a given algorithm $\mathcal{A}$, which functions differently from the membership inference attack used in prior work ([SSSS17, JE19] and described in Appendix D). Here, we describe the procedure generally, in the case where $\delta = 0$; in Appendix A, we show how to adapt the procedure for $\delta > 0$, and in Section 3, we discuss how we instantiate it in our work. Given a learning algorithm $\mathcal{A}$, we construct two datasets $D_0$ and $D_1$ differing on $k$ rows, and some output set $\mathcal{O}$. We defer the discussion of constructing $D_0$, $D_1$, and $\mathcal{O}$ to Section 3. We also wish to bound the probability that we incorrectly measure $\varepsilon_{LB}$ by a small value $\alpha$. From Equation (2), observe that by estimating the quantities $p_0 = \Pr[\mathcal{A}(D_0) \in \mathcal{O}]$ and $p_1 = \Pr[\mathcal{A}(D_1) \in \mathcal{O}]$, we can compute the largest $\varepsilon_{LB}$ such that Equation (2) holds. With $\delta = 0$, this simplifies to $\varepsilon_{LB} = \ln(p_0/p_1)$. This serves as an estimate of the leakage of the private algorithm, but requires estimating $p_0$ and $p_1$ accurately.

For an arbitrary algorithm, it's infeasible to compute $p_0, p_1$ precisely, so we rely on Monte Carlo estimation, by training some fixed $T$ number of times. However, this approach incurs statistical uncertainty, which we correct for by using Clopper Pearson confidence intervals [CP34]. That is, to ensure that our estimate $\varepsilon_{LB}$ holds with probability $> 1 - \alpha$, we find a Clopper Pearson lower bound for $p_1$ that holds with probability $1 - \alpha/2$, and an upper bound for $p_0$ holding with probability $1 - \alpha/2$. Qualitatively, we can be confident that our lower bound on privacy leakage $\varepsilon'$ holds with probability $1 - \alpha$. This procedure is outlined in Algorithm 2, and we prove its correctness in Theorem 2.

---

**Algorithm 2:** Empirically Measuring $\varepsilon$

**Data:** Algorithm $\mathcal{A}$, datasets $D_0, D_1$ at distance $k$, output set $\mathcal{O}$, trial count $T$, confidence level $\alpha$

$\mathrm{ct}_0 = 0, \mathrm{ct}_1 = 0$
**For** $i \in [T]$
    **If** $\mathcal{A}(D_0) \in \mathcal{O}$   $\mathrm{ct}_0 = \mathrm{ct}_0 + 1$
    **If** $\mathcal{A}(D_1) \in \mathcal{O}$   $\mathrm{ct}_1 = \mathrm{ct}_1 + 1$
$\hat{p}_0 =$ CLOPPERPEARSONLOWER$(\mathrm{ct}_0, T, \alpha/2)$
$\hat{p}_1 =$ CLOPPERPEARSONUPPER$(\mathrm{ct}_1, T, \alpha/2)$
**Return** $\varepsilon_{LB} = \ln(\hat{p}_0/\hat{p}_1)/k$

---

**Theorem 2.** When provided with black box access to an algorithm $\mathcal{A}$, two datasets $D_0$ and $D_1$ differing on at most $k$ rows, an output set $\mathcal{O}$, a trial number $T$ and statistical confidence $\alpha$, if Algorithm 2 returns $\varepsilon_{LB}$, then, with probability $1 - \alpha$, $\mathcal{A}$ does not satisfy $\varepsilon'$-DP for any $\varepsilon' < \varepsilon_{LB}$.

We stress that when we say $\varepsilon_{LB}$ is a lower bound with probability $1 - \alpha$, this is only over the randomness of the Monte Carlo sampling, and is not based on any modeling or assumptions. We can always move our confidence closer to 1 by taking $T$ larger.

*Proof of Theorem 2.* First, the guarantee of the Clopper-Pearson confidence intervals is that, with probability at least $1 - \alpha$, $\hat{p}_0 \leq p_0$ and $\hat{p}_1 \geq p_1$, which implies $p_0/p_1 \geq \hat{p}_0/\hat{p}_1$. Second, if $\mathcal{A}$ is

$\varepsilon$-DP, then by group privacy we would have $p_0/p_1 \leq \exp(k\varepsilon)$, meaning $\mathcal{A}$ is *not* $\varepsilon'$-DP for any $\varepsilon' < \frac{1}{k}\ln(p_0/p_1)$. Combining the two statements, $\mathcal{A}$ is *not* $\varepsilon'$ for any $\varepsilon' < \frac{1}{k}\ln(\hat{p}_0/\hat{p}_1) = \varepsilon_{LB}$. $\qquad\square$

The $\varepsilon_{LB}$ reported by Algorithm 2 has two fundamental upper bounds, the provable $\varepsilon_{th}$, and an upper bound, $\varepsilon_{OPT}(T, \alpha)$, imposed by Monte Carlo estimation. The first upper bound is natural: if we run the algorithm on some $\mathcal{A}$ for which the $\varepsilon$ we can prove is $\varepsilon_{th} = 1$, then $\varepsilon_{LB} \leq \varepsilon_{th} = 1$. To understand $\varepsilon_{OPT}(T, \alpha)$, suppose we run 500 trials, and desire $\alpha = 0.01$. The best possible performance is if we get perfect inference accuracy and $k = 1$, where $\text{ct}_0 = 500$ and $\text{ct}_1 = 0$. The Clopper Pearson confidence interval produces $\hat{p}_0 = 0.989, \hat{p}_1 = 0.011$, which gives $\varepsilon_{LB} = 4.54/k = 4.54$. Then, with 99% probability, the true $\varepsilon$ is at least 4.54, and $\varepsilon_{OPT}(T, \alpha) = 4.54$.

We remark that the above procedure only demonstrates that $\mathcal{A}$ cannot be strictly better than $(\varepsilon_{LB}, 0)$-DP, but allows for it to be $(\varepsilon_{LB}/2, \delta)$-DP for very small $\delta$. However, in our work, $\hat{p}_0, \hat{p}_1$ turn out never to be too close to 0, so these differences have little effect on our findings. In Appendix A, we formally discuss how to modify this algorithm for $(\varepsilon, \delta)$-DP for $\delta > 0$. We also show when we can increase $\varepsilon_{LB}$ by considering the maximum upper bounds of the original output set $\mathcal{O}$ and its complement $\mathcal{O}^C$.

# 3 Poisoning Attacks

We now show how to use poisoning attacks to run Algorithm 2. Intuitively, we begin with a dataset $D_0$ and replace $k$ rows with poisoning points to form $D_1$; we then use the impact of poisoning as an output set $\mathcal{O}$. We start with existing backdoor attacks [GDGG17], and then propose a more effective clipping-aware poisoning attack.

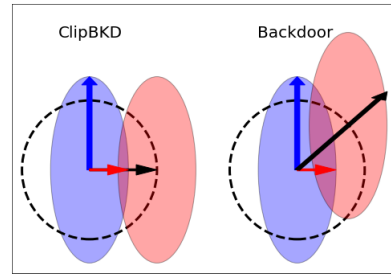

## 3.1 Poisoning Background

In a poisoning attack, an adversary replaces $k$ data points from a training dataset $D$ of $n$ points. The poisoned training dataset is provided as input to the training algorithm, which releases a model $f$ that minimizes a loss function $\mathcal{L}(f, D)$ on its given dataset $D$.

We focus on a specific type of poisoning attack, called a *backdoor attack* [GDGG17]. In a backdoored model, the performance on natural data is maintained, but, by adding a small perturbation to a data point $x$ into $Pert(x)$, the adversary changes the predicted class of the perturbed data. These attacks have been developed for image datasets. In the original attack [GDGG17], described in Algorithm 3,

Figure 1: The distribution of gradients from an iteration of DP-SGD under a clean dataset (blue ellipse) and a poisoned dataset (red ellipse). The right pair depicts traditional backdoors while the left pair depicts our backdoors. Our attack pushes in the direction of least variance, so is impacted less by gradient clipping, which is indicated by the two distributions overlapping less.

the perturbation function $Pert(\cdot)$ adds a pattern in the corner of an image. The poisoning attack takes natural data $(x, y)$, perturbs the image to $Pert(x)$, and changes the class to some $y_p$. The objective is to decrease the loss on $(Pert(x), y_p)$ values from the perturbed test set.

## 3.2 Clipping-Aware Poisoning

DP-SGD makes two modifications to the learning process to preserve privacy: clipping gradients and adding noise. Clipping provides no formal privacy on its own, but many poisoning attacks perform significantly worse in the presence of clipping. Indeed, the basic backdoor attack from Section 3.1 results in a fairly weak lower bound of at most $\varepsilon_{LB} = 0.11$ using the Fashion-MNIST dataset, even with no added noise (which has an $\varepsilon_{th} = \infty$). To improve this number, we must make the poisoning attack sufficiently robust to clipping.

To understand existing backdoor attacks' difficulty with clipping, consider clipping's impact on logistic regression. The gradient of model parameters $w$ with respect to a poisoning point $(x_p, y_p)$ is

$$\nabla_w \ell(w, b; x_p, y_p) = \ell'(w \cdot x_p + b, y_p)x_p.$$

---
**Algorithm 3:** Baseline Backdoor Poisoning Attack and Test Statistic (Section 3.1)
---
**Data:** Dataset $X, Y$, poison size $k$, perturbation function $Pert$, target class $y_p$
**Function** BACKDOOR$(X, Y, k, Pert, y_p)$:
  $\quad X_p = \text{GETRANDOMROWS}(X, k)$
  $\quad X'_p = Pert(X_p)$
  $\quad X^p_{tr} = \text{REPLACERANDOMROWS}(X, X'_p)$
  $\quad Y^p_{tr} = \text{REPLACERANDOMROWS}(Y, y_p)$
  $\quad$**return** $D_0 = (X, Y), D_1 = (X^p_{tr}, Y^p_{tr})$
**Data:** Model $f$, dataset $(X, Y)$, pert. function $Pert$, target class $y_p$, loss function $\ell$, threshold $Z$
**Function** BACKDOORTEST$(f, X, Y, Pert, y_p, \ell, Z)$:
  $\quad X_p = Pert(X)$
  $\quad$**If** $\sum_{x_p \in X_p} \ell(f; (x_p, y_p)) > Z$ **Return** Backdoored
  $\quad$**Return** Not Backdoored
---

---
**Algorithm 4:** Clipping-Aware Backdoor Poisoning Attack and Test Statistic (Section 3.2)
---
**Data:** Dataset $X, Y$, pretrained model $f$, poison size $k$, dataset dimension $d$, norm $m$
**Function** CLIPBKD$(X, Y, k, f, m)$:
  $\quad U, D, V = SVD(X)$ $\qquad\qquad\qquad\qquad\qquad$ $\triangleright$ Singular value decomposition
  $\quad x_p = mV_d$ $\qquad\qquad\qquad$ $\triangleright V_d$ is the singular vector for smallest singular value
  $\quad y_p = \arg\min_i f(x_p)$ $\qquad\qquad\qquad\qquad$ $\triangleright$ Pick class maximizing gradient norm
  $\quad X^p_{tr} = \text{REPLACERANDOMROWS}(X, [x_p] * k)$ $\qquad\quad$ $\triangleright$ Add poisoning point $k$ times
  $\quad Y^p_{tr} = \text{REPLACERANDOMROWS}(Y, [y_p] * k)$ $\qquad\quad$ $\triangleright$ Add targeted class $k$ times
  $\quad$**return** $D_0 = (X, Y), D_1 = (X^p_{tr}, Y^p_{tr})$
**Data:** Model $f$, Poison Data $x_p, y_p$, Threshold $Z$
**Function** CLIPBKDTEST$(f, x_p, y_p, Z)$:
  $\quad$**If** $(f(x_p) - f(0^d)) \cdot y_p > Z$ **Return** Backdoored
  $\quad$**Return** Not Backdoored
---

Standard poisoning attacks, including the backdoor attack from Section 3.1, focus on increasing $|\ell'(w \cdot x_p + b, y_p)|$; by doubling this quantity, if $|x_p|$ is fixed, half as many poisoning points are required for the same effect. However, in the presence of clipping, this relationship is broken.

To be more effective in the presence of clipping, the attack must produce not only large gradients, but *distinguishable* gradients. That is, the distribution of gradients arising from poisoned and cleaned data must be significantly different. To analyze distinguishability, we consider the variance of gradients, illustrated in Figure 1, and seek a poisoning point $(x_p, y_p)$ minimizing $Var_{(x,y) \in D}[\nabla_w \ell(w, b; x_p, y_p) \cdot \nabla_w \ell(w, b; x, y)]$. This is dependent on the model parameters at a specific iteration of DP-SGD: we circumvent this issue by minimizing the following upper bound, which holds for all models (for logistic regression, $|\ell'(w \cdot x + b; y)| \leq 1$):

$$Var_{(x,y) \in D}[\ell'(w \cdot x_p + b, y_p)x_p \cdot \ell'(w \cdot x + b, y)x] \leq Var_{(x,y) \in D}[x_p \cdot x].$$

We can minimize this variance with respect to the poisoning point $x_p$ by using the singular value decomposition: selecting $x_p$ as the singular vector corresponding to the smallest singular value (i.e. the direction of least variance), and scale $x_p$ to a similar norm to the rest of the dataset. We select $y_p$ to be the smallest probability class on $x_p$. We then insert $k$ copies of the poisoning point $(x_p, y_p)$. We call this approach CLIPBKD, detailed in Algorithm 4. We prove in Appendix B that when we run CLIPBKD (modified for linear regression) to estimate the privacy of the *output perturbation* algorithm, we obtain $\varepsilon_{LB}$ within a small factor of the upper bound $\varepsilon_{th}$, giving evidence that this attack is well suited to our application in differential privacy. In addition, our theoretical analysis highlights a data-dependence—the attack is more effective when the data has directions of small variance. As a result, it may be that more "spherically distributed" datasets are less vulnerable to privacy violations. In Appendix C, we describe how to adapt CLIPBKD to transfer learning from a pre-trained model.

For both standard and clipping-aware backdoors, we generate $D_0, D_1$ with a given poisoning size $k$, using functions BACKDOOR or CLIPBKD, respectively. With larger $k$, the poisoning attack will be

| Dataset | Epochs | Learning Rate | Batch Size | $\ell_2$ Regularization |
|---------|--------|---------------|------------|--------------------------|
| FMNIST | 24 | 0.15 | 250 | 0 |
| CIFAR10 | 20 | 0.8 | 500 | 0 |
| P100 | 100 | 2 | 250 | $10^{-4}$ / $10^{-5}$ |

Table 1: Training details for experiments in Section 4. P100 regularization is $10^{-5}$ for logistic regression and $10^{-4}$ for neural networks, following [JE19].

more distinguishable, making it more effective when there is more noise (when $\varepsilon_{th}$ is small). Then the test statistic is whether the backdoored points are distinguishable by a threshold on their loss (i.e., output set $\mathcal{O}$ is whether BKDTEST or CLIPBKDTEST return "Backdoored"). We first run an initial phase of $T$ trials to find a good threshold $Z$ for the test functions. We then run another $T$ trials in Algorithm 2 to estimate $\hat{p_0}$ and $\hat{p_1}$ based on either the BKDTEST or the CLIPBKDTEST test statistic.

## 4    Experiments and Discussion

### 4.1    Experimental Setup

We evaluate both membership inference (MI, as used by [YGFJ18] and [JE19] and described in Appendix D) and our algorithms on three datasets: Fashion-MNIST (FMNIST), CIFAR10, and Purchase-100 (P100). For each dataset, we consider both a **logistic regression (LR)** model and a **two-layer feedforward neural network (FNN)**, trained with DP-SGD using various hyperparameters:

**FMNIST** [XRV17] is a dataset of 70000 28x28 pixel images of clothing from one of 10 classes, split into a train set of 60000 images and a test set of 10000 images. It is a standard benchmark dataset for differentially private machine learning. To improve training speed, we consider a simplified version, using only classes 0 and 1 (T-shirt and trouser), and downsample so each class contains 3000 training and 1000 testing points. **CIFAR10** [Kri09] is a harder dataset than FMNIST, consisting of 60000 32x32x3 images of vehicles and animals, split into a train set of 50000 and a test set of 10000. For training speed, we again take only class 0 and 1 (airplane and automobile), making our train set contain 10000 samples, and the test set 2000 samples. For models we train on CIFAR10, we follow standard practice for differential privacy and fine-tune the last layer of a model pretrained nonprivately on the more complex CIFAR100, a similarly sized but more complex benchmark dataset [PCS+20]. **P100** [SSSS17] is a modification of a Kaggle dataset [Pur], with 200000 records of 100 features, and 100 classes. The features are purchases, and the classes are user clusters. Following [JE19], we subsample the dataset so it has 10000 train records and 10000 test records.

**Model Size.**    The two-layer feedforward neural networks all have a width of 32 neurons. For CIFAR10, the logistic regression model and feedforward neural network are added on top of the pretrained convolutional neural network.

**Computing Thresholds.**    In order to run Algorithm 2, we need to specify $D_0, D_1$ and $\mathcal{O}$. We've described how to use poisoning to compute $D_0, D_1$, and how the test statistics for these attacks are constructed, assuming a known threshold. To produce this threshold, we train 500 models on the unpoisoned dataset and 500 models on the poisoned dataset, and identify which of the resulting 1000 thresholds produces the best $\varepsilon_{LB}$, using Algorithm 2.

**Training Details.**    We discuss the details of training in Table 1. We selected these values to ensure a good tradeoff between accuracy and $\varepsilon$, and selecting $\ell_2$ regularization for P100 based on [JE19].

Our techniques are general, and could be applied to any dataset-model pair to identify privacy risks for DP-SGD. Examining these six dataset-model pairs demonstrates that our technique can be used to identify new privacy risks in DP-SGD, and a comprehensive empirical study is not our focus.

### 4.2    Results and Discussion

Figure 2 presents a direct comparison of the privacy bounds produced by ClipBKD (our attack), the standard backdoor attack, and MI. As standard backdoor attacks only exist for images, we only report results on them on FMNIST and CIFAR10. The pattern we choose for backdoor attacks is a 5x5 white square in the top-left corner of an image. For ClipBKD, we use $T = 500$ trials and confidence

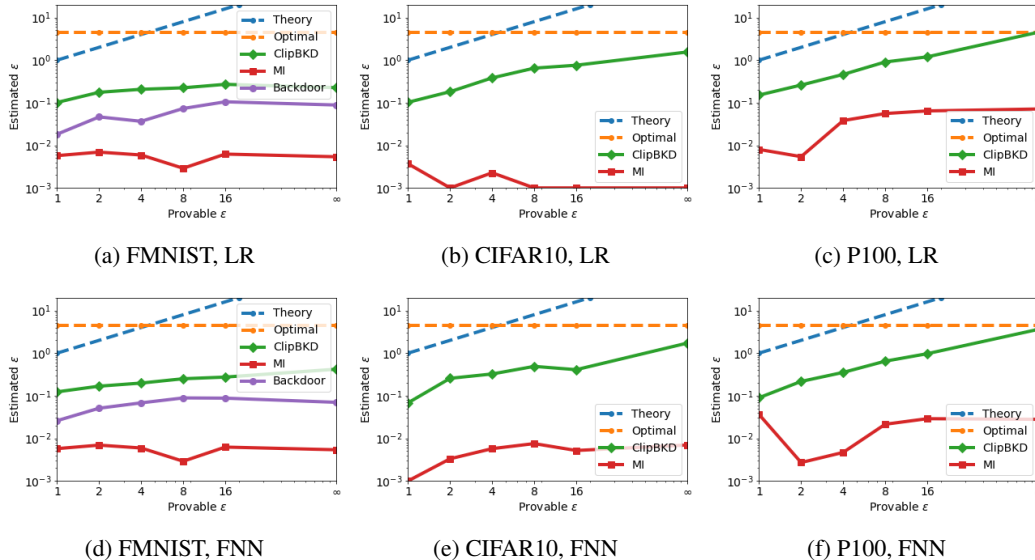

Figure 2: Performance of privacy attacks MI (Membership Inference—[YGFJ18, JE19]), Backdoor, and ClipBKD on our datasets. LR = logistic regression, FNN = two-layer neural network. Backdoor attacks have not been developed for Purchase-100, so only MI and Clip-BKD were run. Backdoors do not provide positive $\varepsilon_{LB}$ on CIFAR10 due to difficulty with the pretrained model.

level $\alpha = 0.01$ (i.e., our Monte Carlo estimates hold with 99% confidence) and report the best result from $k = 1, 2, 4, 8$ poisoning points. Results for MI use 1000 samples, and average over 10 trained models. For context, we display the best theoretical upper bound on $\varepsilon_{th}$ and also $\varepsilon_{OPT}(500, 0.01)$, which is the best value of $\varepsilon_{LB}$ that we could hope to produce using $T$ trials and confidence level $\alpha$.

For every dataset and model, we find that ClipBKD significantly outperforms MI, by a factor of between 2.5x and 1500x. As a representative example, for $\varepsilon_{th} = 4$ on Purchase-100 with 2-layer neural networks, ClipBKD gives an $\varepsilon_{LB}$ of 0.46, while MI gives $\varepsilon_{LB}$ of 0.04, an improvement of 12.1x. We also find ClipBKD always improves over standard backdoors: on FMNIST by an average factor of 3.84x, and standard backdoors never reach positive $\varepsilon_{LB}$ on CIFAR, due to the large number of points required to poison the pretrained model. We also find that ClipBKD returns $\varepsilon_{LB}$ that are close to $\varepsilon_{th}$; for finite $\varepsilon_{th}$, the majority of gaps are a factor of $< 12.3$x, reaching as low as 6.6x. For example, on Purchase-100, when $\varepsilon_{th} = 4$, we find that ClipBKD returns an $\varepsilon_{LB}$ of 0.46, a gap of 8.7x.

**Sensitivity to Hyperparameters.** We also give a more thorough evaluation of ClipBKD's performance as a function of DP-SGD's hyperparameters. We vary clipping norm between 0.5, 1, and 2 and vary the noise to ensure $\varepsilon_{th}$ between 1, 2, 4, 8, 16, and $\infty$. We also vary the initialization randomness between random normal initializations with variance 0 (fixed initialization), $0.5\sigma$, $\sigma$, and $2\sigma$, where $\sigma$ is the variance of Glorot normal initialization. Table 2 reports the best $\varepsilon_{LB}$ produced by the attack over $k = 1, 2, 4, 8$. Our best measured values of $\varepsilon_{LB}$ occur when initialization is fixed, and are within a 4.2-7.7x factor of $\varepsilon_{th}$, speaking to the effectiveness of ClipBKD. When $\varepsilon_{th} = \infty$ and the initialization is fixed, we achieve perfect inference accuracy, matching $\varepsilon_{OPT}(500, 0.01) = 4.54$.

These experiments reveal three intuitive trends. First, as $\varepsilon_{th}$ increases (equivalently, the noise level decreases), $\varepsilon_{LB}$ also increases. Second, as the initialization randomness decreases, $\varepsilon_{LB}$ increases. All existing analyses of DP-SGD give privacy upper bounds for *any fixed initialization*, and our results suggest that initial randomization might play a significant role. Finally, as clipping norm decreases, $\varepsilon_{LB}$ decreases, except when the initialization is fixed. In fact, our results show that $\varepsilon_{LB}$ is more sensitive to the clipping norm than the amount of noise. All existing analyses of DP-SGD consider only the *noise multiplier* $\sigma_{GD}$ but not the clipping norm, but the role of the clipping norm itself seems highly significant.

| Params | Fixed Init | Init Rand = $0.5\sigma$ | Init Rand = $\sigma$ | Init Rand = $2\sigma$ |
|---|---|---|---|---|
| $\varepsilon_{th} = 1, \sigma_{GD} = 5.02$ | 0.13 / 0.15 / 0.13 | 0.13 / 0.17 / 0.13 | 0.06 / 0.12 / 0.09 | 0.01 / 0.06 / 0.08 |
| $\varepsilon_{th} = 2, \sigma_{GD} = 2.68$ | 0.33 / 0.37 / 0.28 | 0.27 / 0.33 / 0.39 | 0.10 / 0.17 / 0.27 | 0.01 / 0.06 / 0.17 |
| $\varepsilon_{th} = 4, \sigma_{GD} = 1.55$ | 0.89 / 0.75 / 0.71 | 0.28 / 0.52 / 0.78 | 0.08 / 0.20 / 0.54 | 0.02 / 0.10 / 0.18 |
| $\varepsilon_{th} = 8, \sigma_{GD} = 1.01$ | 1.61 / 1.85 / 1.90 | 0.33 / 0.55 / 1.27 | 0.07 / 0.25 / 0.53 | 0.01 / 0.05 / 0.20 |
| $\varepsilon_{th} = 16, \sigma_{GD} = 0.73$ | 2.15 / 2.16 / 2.43 | 0.36 / 0.80 / 1.39 | 0.13 / 0.27 / 0.72 | 0.02 / 0.08 / 0.16 |
| $\varepsilon_{th} = \infty, \sigma_{GD} = 0$ | 4.54 / 4.54 / 4.54 | 0.29 / 0.95 / 2.36 | 0.10 / 0.42 / 0.79 | 0.03 / 0.09 / 0.27 |

Table 2: Lower bound $\varepsilon_{LB}$ measured with CLIPBKD for clipping norms of (0.5 / 1 / 2) for two-layer neural networks trained on FMNIST. Training accuracy for all models is 96%-98%. Results are the maximum over $k = 1, 2, 4, 8$. $\sigma_{GD}$ refers to the DP-SGD noise multiplier, while $\sigma$ is Glorot initialization randomness [GB10]. All reported values of $\varepsilon_{LB}$ are valid with 99% confidence over the randomness of our experiments.

| Params | Init Rand = $0.5\sigma$ | Init Rand = $\sigma$ | Init Rand = $2\sigma$ |
|---|---|---|---|
| $\varepsilon_{th} = 1, \sigma_{GD} = 7.78$ | 0.09 / 0.01 / 0.00 | 0.05 / 0.00 / 0.00 | 0.07 / 0.05 / 0.00 |
| $\varepsilon_{th} = 2, \sigma_{GD} = 4.04$ | 0.16 / 0.27 / 0.11 | 0.21 / 0.17 / 0.03 | 0.20 / 0.10 / 0.02 |
| $\varepsilon_{th} = 4, \sigma_{GD} = 2.20$ | 0.38 / 0.33 / 0.30 | 0.29 / 0.35 / 0.30 | 0.34 / 0.33 / 0.13 |
| $\varepsilon_{th} = 8, \sigma_{GD} = 1.31$ | 0.52 / 0.53 / 0.42 | 0.54 / 0.53 / 0.52 | 0.56 / 0.46 / 0.50 |
| $\varepsilon_{th} = 16, \sigma_{GD} = 0.89$ | 0.80 / 0.77 / 0.71 | 0.63 / 0.77 / 0.76 | 0.74 / 0.70 / 0.72 |
| $\varepsilon_{th} = \infty, \sigma_{GD} = 0$ | 2.73 / 4.53 / 4.54 | 1.52 / 3.08 / 4.52 | 0.90 / 1.91 / 2.79 |

Table 3: Lower bound $\varepsilon_{LB}$ measured with CLIPBKD for clipping norms of (0.5 / 1 / 2) for two-layer neural networks trained on P100. Results are the maximum over $k = 1, 2, 4, 8$. $\sigma_{GD}$ refers to the DP-SGD noise multiplier, while $\sigma$ is Glorot initialization randomness [GB10]. We do not run experiments with fixed initialization as we already achieve $\varepsilon_{OPT}(500, 0.01)$ with initialization of $0.5\sigma$. All reported values of $\varepsilon_{LB}$ are valid with 99% confidence over the randomness of our experiments.

We emphasize that for *every* choice of the hyperparameters, the training accuracy is 96–98%, so the algorithm has comparable utility, but potentially very different privacy and robustness to poisoning, as we vary these parameters. We believe these phenomena deserve further analytical study.

On the P100 dataset, we perform an identical parameter sweep, shown in Table 3. On P100, we use $\ell_2$ regularization, a higher learning rate, and more epochs, making the contribution of the initialization smaller. As we would expect from these modifications, the role of both clipping norm and random initialization are diminished.

## 5   Conclusion and Future Directions

In this work we use novel poisoning attacks to establish strong limits on the privacy of specific differentially private algorithms, namely DP-SGD. We establish that the worst-case privacy bounds for this algorithm are approaching their limits.

Our findings highlight several questions for future exploration: (1) How much can our attacks be pushed quantitatively? Can the gap between our lower bounds and the worst-case upper bounds be closed? (2) Can we incorporate additional features into the privacy analysis of DP-SGD, such as the specific gradient-clipping norm, and the amount of initial randomness? (3) How realistic are the instances produced by our attacks, and can we extend the attacks to give easily interpretable examples of privacy risks for non-experts?

Although there is no hope of determining the precise privacy level of a given algorithm in a fully empirical way, we believe our work demonstrates how a quantitative, empirical approach to privacy attacks can effectively complement analytical work on privacy in machine learning.

## 6   Broader Impact

Differentially private algorithms have the potential to unlock the societal benefits of analyzing datasets of sensitive information while giving strong protections of *individual privacy*. While all differentially private algorithms offer a strong *qualitative* privacy guarantee, the specific *quantitative* implications

of specific deployments are not as well understood, and, in order to give high utility, these systems only offer weak worst-case guarantees. The concern is that such systems *might* only be offering a veneer of individual privacy, and we believe this status quo poses significant risk. We believe our work is a step towards giving organizations the information and tools they need to make informed decisions about the implications of specific deployments of differential privacy. Moreover, our work can support those working on new methods and tools to ultimately push the Pareto frontier of privacy-utility tradeoffs forward.

We note that the sort of individual privacy guarantees do not address all possible concerns about how data is used, not even all concerns that are informally described as "privacy concerns." Our work cannot ultimately answer when and how differential privacy should or should not be used in specific applications, it merely provides the people entrusted with making those decisions with correct information. This situation is similar to cryptography, where tools like encryption address specific problems, and must be used carefully and only when appropriate, but better encryption technology, or exposing the flaws of specific encryption technologies enables better policy decisions.

Lastly, we point out that our work only exposes *potential* privacy risks in experiments based on *public* benchmark datasets. We have not used our methods to reveal any sensitive information about real individuals that was previously believed to be private. As with cryptanalysis, it is important to follow ethical disclosure guidelines of any attacks against deployed systems to balance the value of public knowledge of the breach versus harm to existing users.

## 7  Funding Disclosure

JU is supported by NSF grants CCF-1750640, CNS-1816028, and CNS-1916020, and a Google Faculty Research Award. This research was also sponsored by the U.S. Army Combat Capabilities Development Command Army Research Laboratory under Cooperative Agreement Number W911NF-13-2-0045 (ARL Cyber Security CRA). The views and conclusions contained in this document are those of the authors and should not be interpreted as representing the official policies, either expressed or implied, of the Combat Capabilities Development Command Army Research Laboratory or the U.S. Government. The U.S. Government is authorized to reproduce and distribute reprints for Government purposes notwithstanding any copyright notation here on.

## Footnotes

[1]There are several common variants of differential privacy [DKM+06, DR16, BS16, Mir17, BDRS18, DRS19] that quantify the influence of a single example in slightly different ways, sometimes using more than one parameter. For this high-level discussion, we focus on the single, primary parameter $\varepsilon$.

[2]Although our methods are general, in this work we exclusively study the implementation and privacy analysis of DP-SGD in TensorFlow Privacy [Goo].

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
