[Supplementary Material]

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

    $X_p = $ GETRANDOMROWS($X, k$)
    $X'_p = Pert(X_p)$
    $X^p_{tr} = $ REPLACERANDOMROWS($X, X'_p$)
    $Y^p_{tr} = $ REPLACERANDOMROWS($Y, y_p$)
    **return** $D_0 = (X, Y), D_1 = (X^p_{tr}, Y^p_{tr})$
**Data:** Model $f$, dataset $(X, Y)$, pert. function $Pert$, target class $y_p$, loss function $\ell$, threshold $Z$
**Function** BACKDOORTEST($f, X, Y, Pert, y_p, \ell, Z$):
    $X_p = Pert(X)$
    **If** $\sum_{x_p \in X_p} \ell(f; (x_p, y_p)) > Z$ **Return** Backdoored
    **Return** Not Backdoored

---

**Algorithm 4:** Clipping-Aware Backdoor Poisoning Attack and Test Statistic (Section 3.2)

---

**Data:** Dataset $X, Y$, pretrained model $f$, poison size $k$, dataset dimension $d$, norm $m$
**Function** CLIPBKD($X, Y, k, f, m$):
    $U, D, V = SVD(X)$                                           ▷ Singular value decomposition
    $x_p = mV_d$                     ▷ $V_d$ is the singular vector for smallest singular value
    $y_p = \arg\min_i f(x_p)$              ▷ Pick class maximizing gradient norm
    $X^p_{tr} = $ REPLACERANDOMROWS($X, [x_p] * k$)     ▷ Add poisoning point $k$ times
    $Y^p_{tr} = $ REPLACERANDOMROWS($Y, [y_p] * k$)     ▷ Add targeted class $k$ times

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

# A  Extending Algorithm 2.

**Measuring $\varepsilon$ when $\delta > 0$.** Notice that Algorithm 2 holds for $(\varepsilon, 0)$-differential privacy. However, this is only for simplicity—the group privacy guarantee of $(\varepsilon, \delta)$-differential privacy can be converted to a similar procedure. Write $x = \exp(\varepsilon)$ in the group privacy guarantee for Equation 2, and rearrange to provide $p_1 x^{k+1} - (p_1 - \delta)x^k - p_0 x + (p_0 - \delta) \geq 0$. We can solve for $x$ here using a root-finding algorithm to find $x$, and computing $\varepsilon_{LB} = \ln(x)$. Theorem 2 can be easily extended to this case.

Figure 3: $f_0(x)$ and $f_1(x)$, as defined in Lemma 3 with $\delta = 10^{-5}$, $k = 4$, $p_0 = 0.6$, $p_1 = 0.8$.

**Measuring $\varepsilon_{LB}$ with both $\mathcal{O}$ and $\mathcal{O}^C$.** Notice that differential privacy makes a guarantee for all output sets, including the complement $\mathcal{O}^C$; if $\Pr[A(D) \in \mathcal{O}] = p$, then $\Pr[A(D) \in \mathcal{O}^C] = 1 - p$. If, upon computing $p_0, p_1$, we can compute a larger $\varepsilon_{LB}$ by using $\mathcal{O}^C$, this requires no extra trials.

For example, suppose $\delta = 0$, $k = 1$, $p_1 = 0.8$, and $p_0 = 0.4$. Here, $\varepsilon_{LB} = \ln(p_1/p_0)/1 = \ln(2)$. If, instead, we replace $\mathcal{O}$ by $\mathcal{O}^C$, $\varepsilon_{LB} = \ln((1 - p_0)/(1 - p_1))/1 = \ln(0.6/0.2) = \ln(3)$. In Lemma 3, we show when this technique improves $\varepsilon_{LB}$: when $p_1 > p_0 + k\delta$ and $p_0 + p_1 > 1$. We use this modification in all of our experiments.

**Lemma 3.** If $p_1 > p_0 + k\delta$ and $p_0 + p_1 > 1$, then the largest root of $f_0(x) = p_1 x^{k+1} - (p_1 - \delta)x^k - p_0 x + (p_0 - \delta)$ is smaller than the largest root of $f_1(x) = (1 - p_0)x^{k+1} - (1 - p_0 - \delta)x^k - (1 - p_1)x + (1 - p_1 - \delta)$.

*Proof.* Write $x_0$ the largest root of $f_0(x)$, and $x_1$ the largest root of $f_1(x)$. We show this in two parts: first, we show that, for all $p_0, p_1$, $f_0(x)$ has a root $x > 1$ when $p_1 > p_0 + k\delta$, after which it is monotonically increasing. This shape is evident in Figure 3. This provides a nonzero $\varepsilon_{LB}$. Then we show that this $\varepsilon_{LB}$ must be smaller for $f_1(x)$ when $p_0 + p_1 > 1$, because $f_0(x) - f_1(x) > 0$ when $x > 1$. This ensures that $f_1(x_0) < 0$, and so the root $x_1 > x_0$.

We begin by showing that $f_0(x)$ has a single root $x > 1$. First, notice that $f_0(1) = 0$. We then analyze the derivative, showing that it starts negative, has a root, and then is always positive. This indicates that there can only be one root.

$$f_0'(x) = (k+1)x^k p_0 + kx^{k-1}(\delta - p_0) - p_1 = kp_0 x^{k-1}(x-1) + x^k p_0 + k\delta x^{k-1} - p_1$$

This has a root $x > 1$ if $x^k p_0 + k\delta x^{k-1} - p_1 < 0$, so we require $p_0 + k\delta - p_1 < 0$. Notice too that $f_0'(x)$ is monotonically increasing at $x > 1$. This ensures that it has only one root $x > 1$. This argument holds, too, for $f_1(x)$, as if $p_0 + k\delta - p_1 < 0$, then $(1 - p_1) + k\delta - (1 - p_0) < 0$.

Now that we know both $f_0$ and $f_1$ only have a single root, and they are both increasing at that root, we just need to show that $f_1(x_0) < 0$, as this will ensure $x_1 > x_0$. We do this by showing that $\forall x > 1$, $f_0(x) - f_1(x) > 0$. First, write

$$f_0(x) - f_1(x) = x^{k+1}(p_0 + p_1 - 1) + x^k(1 - p_0 - p_1) + x(1 - p_0 - p_1) + p_0 + p_1 - 1.$$

The $\delta$ terms cancel, and we can factor the above into

$$f_0(x) - f_1(x) = (p_0 + p_1 - 1)(x - 1)(x^k - 1).$$

This is always positive when $x > 1$ and $p_0 + p_1 > 1$.  $\square$

# B  Analysis of Backdoor Poisoning-based Auditing

We now provide formal evidence for the effectiveness of backdoor poisoning attacks in auditing differentially private algorithms with a case study on linear regression. To the best of our knowledge, this is also the first formal analysis of backdoor poisoning attacks for a concrete learning algorithm.

**Theorem 4.** Given a dataset $X \in \mathbb{R}^{n \times d}, Y \in [-.5, .5]^n$, where each $x_i \in X$ satisfies $|x_i|_2 \leq 1$, consider output perturbation to compute ridge regression with regularization $\lambda$, satisfying $\varepsilon, \delta$ differential privacy. Then Algorithm 5 produces a backdoor attack that produces a lower bound

$$\varepsilon_{LB} = \frac{\lambda \varepsilon}{(1 + \lambda + \sigma_d)\sqrt{\pi \ln(1.25/\delta)}} - 4\delta$$

where $\sigma_d$ is the smallest singular value of $X$.

---

**Algorithm 5:** Clipping-Aware Poisoning Attack Generation

---

**Data:** Dataset $X \in \mathbb{R}^{n \times d}, Y \in [-.5, .5]$
**Result:** $D_c = (X_p, Y_p), D_p = (X_p, Y_p')$
$U, D, V = SVD(X)$                     $\triangleright$ Singular value decomposition
$x_p = V_d$
$y_p = .5$
**Return** $(X \parallel x_p, Y \parallel y_p), (X \parallel x_p, Y \parallel -y_p)$

---

*Proof.* We begin by computing the difference between the optimal linear regression parameters $w_0, w_1$ for the two datasets $D_0 = (X \parallel x_p, Y \parallel y_p), D_1 = (X \parallel x_p, Y \parallel -y_p)$, respectively. We refer to $X_p = (X \parallel x_p), Y_0 = (Y \parallel y_p), Y_1 = (Y \parallel -y_p)$. We continue to refer to the eigendecomposition of $X^T X = VDV^T$. Recall that the optimal parameters for an arbitrary dataset $X, Y$ with $\ell_2$ regularization $\lambda$ is $(\lambda I + X^T X)^{-1} X^T Y$.

$$
\begin{aligned}
w_0 - w_1 &= (\lambda I + X_p^T X_p)^{-1} X_p^T (Y_0 - Y_1) \\
&= (\lambda I + X_p^T X_p)^{-1} x_p \\
&= (\lambda I + VDV^T + v_d v_d^T)^{-1} v_d \\
&= (\lambda V I V^T + VDV^T + V\mathrm{diag}(e_d)V^T)^{-1} v_d \\
&= V(\lambda I + D + \mathrm{diag}(e_d))^{-1} V^T v_d \\
&= \frac{v_d}{\lambda + \sigma_d + 1}
\end{aligned}
$$

The output perturbation algorithm for Ridge regression, with $(\varepsilon, \delta)$-DP, adds Gaussian noise with variance $\sigma^2 = 2\ln(1.25/\delta)(2/\lambda)^2/\varepsilon^2$ to the optimal parameters $w$.

The optimal distinguisher for $w_0 + \mathcal{N}(0, \sigma^2 \mathbb{I})$ and $w_1 + \mathcal{N}(0, \sigma^2 \mathbb{I})$ is
$$f(w) = \mathbb{1}[w \cdot v_d - 0.5(w_0 + w_1) \cdot v_d < 0].$$
Letting $c = \frac{0.5}{\lambda + \sigma_d + 1}$, the probability of success for this distinguisher is $\Pr[\mathcal{N}(0, \sigma^2) < c]$, which gives an $\varepsilon$ lower bound of
$$\ln\left(\frac{\Pr[\mathcal{N}(0, \sigma^2) < c] - \delta}{\Pr[\mathcal{N}(0, \sigma^2) < -c]}\right).$$
We can lower bound $\Pr[0 < \mathcal{N}(0, \sigma^2) < c]$ using the following integral approximation:
$$\Pr[0 < \mathcal{N}(0, \sigma^2) < c] \geq \frac{c}{\sigma\sqrt{2\pi}} \exp(-c^2/2\sigma^2),$$
so our $\varepsilon$ lower bound is
$$\ln\left(\frac{0.5 - \delta + c/(\sigma \exp(c^2/2\sigma^2)\sqrt{2\pi})}{0.5 - c/(\sigma \exp(c^2/2\sigma^2)\sqrt{2\pi})}\right) \geq \ln\left(\frac{0.5 - \delta + c/(\sigma \exp(c^2/2\sigma^2)\sqrt{2\pi})}{0.5 + \delta - c/(\sigma \exp(c^2/2\sigma^2)\sqrt{2\pi})}\right).$$
By its Maclaurin series, $\ln\left(\frac{0.5+x}{0.5-x}\right) \geq 4x$. Then we can compute our lower bound on $\varepsilon$ to be

$$\varepsilon_{LB} \geq 4\left(\frac{c}{\sigma \exp(c^2/2\sigma^2)\sqrt{2\pi}} - \delta\right) = O\left(\frac{\lambda \varepsilon}{(1 + \lambda + \sigma_d)\sqrt{\ln(1/\delta)}}\right)$$

so the attack differs by a constant factor from the provable $\varepsilon$.

$\square$

## C   ClipBKD with Pretrained Models

State-of-the-art differentially private CIFAR10 models use transfer learning from a fixed pretrained CIFAR100 convolutional neural network. We call the pretrained model function $f_0$, which is never updated during training. Training produces a $f_1$, such that the entire prediction model is $f(x) = f_1(f_0(x))$.

This makes ClipBKD not directly applicable, as ClipBKD requires access to the input of the trained model $f_1$. Then we must try to produce some $x$ such that $f_0(x) = h_p$, where $h_p$ is produced by SVD in Algorithm 3. This is not in general possible, so we instead use gradient descent to optimize the combination of two loss functions on $x$.

Our first loss function incentivizes decreasing $h_p \cdot v$ for high-variance directions $v \in V_{high}$ from SVD. This ensures the gradient will not move in SGD's noisy directions. Our second loss function incentivizes increasing $h_p \cdot v$ for low-variance directions $v \in V_{low}$ from SVD, ensuring the gradient is distinguishable in low variance directions. Putting these together, we produce $x_p$ by optimizing the following loss function:

$$x_p = \arg\max_x \sum_{v \in V_{low}} (f_0(x) \cdot v)^2 - \sum_{v \in V_{high}} (f_0(x) \cdot v)^2$$
$$\text{s.t. } x \in [0, 1]^d \tag{3}$$

We perform this optimization by projected gradient descent, running 10000 iterations with a learning rate of 1.

## D   Membership Inference

In membership inference [YGFJ18], an adversary is given a model $f$ and its training loss $c$ and seeks to understand whether a given data point $(x, y)$ was used to train the model. Although alternative formulations have been proposed [SSSS17], we focus on the one proposed by [YGFJ18]. Intuitively, the attack relies on a generalization gap: the loss on training data should be smaller than the loss on test data. The algorithm is provided a set of $2n$ elements, $n$ of which were used for training, and $n$ not used for training, and predicts that any sample with loss lower than the training loss. The accuracy of these predictions is bounded by $\frac{\exp(\varepsilon)}{1+\exp(\varepsilon)}$ for any $\varepsilon$-differentially private algorithm, so we can produce a lower bound $\varepsilon_{LB}$ from it. The algorithm is provided in Algorithm 6.

---

**Algorithm 6:** Membership Inference [YGFJ18]

---

**Data:** Training dataset $D_{tr}$ of size $n$, Test dataset $D_t$ of size $n$, Dataset, learning algorithm $\mathcal{A}$

$f, c = \mathcal{A}(D_{tr})$         $\triangleright$ $\mathcal{A}$ returns model and training loss
correct_ct $= 0$
**For** $(x, y) \in D_{tr}$
  |   **If** $\ell(f; x, y) < c$ correct_ct $=$ correct_ct $+ 1$       $\triangleright$ training set should have small loss;
**For** $(x, y) \in D_t$
  |   **If** $\ell(f; x, y) > c$ correct_ct $=$ correct_ct $+ 1$       $\triangleright$ test set should have higher loss;
Adv $=$ correct_ct$/2n$
$\varepsilon_{LB} = \ln(\text{Adv}/(1 - \text{Adv}))$

---