[Reviews · NeurIPS 2020]

Review 1

Summary and Contributions: The paper studies the privacy guarantees of private stochastic gradient descent based method. These methods are prevalently used in privacy preserving machine learning in practice. One key idea is to clip the gradient computed at every iteration so that they have bounded norm, and then add Gaussian noise to the clipped gradient. To perform this in practice, often the choice of privacy parameters become very important. It is common to pick epsilon greater than 1 in practice while usually in theory, epsilon is assumed to be smaller than 1. Higher epsilon means less privacy and lower epsilon means higher privacy. However, in practice, there is a large gap between strong formal guarantees promised by DP and implications of choice of the privacy parameter in practice. There has been very few work in this domain, most notably by Jayaram and Evans, and Carlini et al. The current submission improves their result quantitatively by a factor of 10.

Strengths: The current submission uses data poisoning attack that is more robust to gradient clipping. The key observation in the paper is that previous works poisoned the data randomly and it is possible that the effect of these poisoning data would be nullified if they are in the direction where the gradient was large enough (and hence get clipped). The paper uses this observation and finds the direction in which the variance of the gradient is low and launch poisoning attack accordingly. In more details, the feature vector is the singular vector of the lowest singular value of the variance. The label is then the class that maximises the gradient norm. The paper then add random k copies of these poisoned data. I like the central idea in the paper and I believe it makes a significant improvement in the direction that has a lot of practical relevance.

Weaknesses: My biggest concern with this work is that I believe they oversell their result. Their variance based computation of singular vectors (and hence data poisoning attack) relies heavily on the fact that we can have a good understanding of variance, which is model dependence. It is easiest for logistic regression. I suspect that is the reason the paper looked at logistic regression. As the bound before the line 225 is not tight for many other learning task, I doubt that they would have such a large improvement. That is why I feel the paper oversell their result by showing only where they can show quantifiable improvement. I suspect that the central idea needs more fleshing out to generalise it to other class of learning problem. Post rebuttal: I thank the authors for clarifying this misunderstanding on my part. I think they have answered my question to a reasonable degree. My other concern is the audience. This is an attack paper and I feel a much better audience for the paper would be Usenix, CCS, or Oakland. It would have a much wider audience there. Post rebuttal: I still believe that Usenix, CCS, or Oakland would have got more audience for the paper.

Correctness: Yes. Why do we see a dip in the membership inference attack estimated epsilon when provable epsilon is in some range (depending on the dataset being used). Postrebuttal: It is very strange that there is so much variance even running the experiment few times. My intuition was when provable epsilon tends to infinity, the clipBKD should tend to meet the optimal value. I do not see a reason for doing clipping when we are dealing with epsilon = \infty. Does epsilon=\infty means that we do not do any clipping at all? Post rebuttal: The response are fair enough.

Clarity: Yes, it is well written.

Relation to Prior Work: Yes, it is clearly discussed, except that they miss out a result by Carlini et al. (Usenix).

Reproducibility: Yes

Additional Feedback: What does it mean y_p to be the smallest probability class on x_p? What is probability class? Post rebuttal: thanks for the clarification. I have a question to the authors, which I think would help the readers get a better understanding: why do we need to add multiple copies of the same poisoning data. Why not we estimate the least k singular vectors and use different poisoning data corresponding to these vectors. Post rebuttal: thanks for the clarification.


Review 2

Summary and Contributions: This paper performs study on the privacy of DP-SGD in practice. Quantitative and empirical approach are adopted for the study, interesting findings and insights are discussed by evaluation.

Strengths: + This paper works on an interesting and problem. + A new poisoning attack is proposed + An empirical method is proposed for privacy estimation, which could be a complement of analytical DP.

Weaknesses: - This paper mainly focuses on the privacy of SGD. It is unclear whether it generalizes to other cases - The evaluation is only performed on 3 small datasets and small models, unclear whether it can be applicable to practical datasets and models. One claim of this paper is to complement analytical estimation, whether the proposed technique scales in practice is a concern. - In addition, it is unclear either whether the proposed methods support more complicated DNN architectures such as recurrent neural networks and transformers.

Correctness: The claims and methods are feasible, and properly evaluated.

Clarity: The paper is well written and easy to follow

Relation to Prior Work: Largely.

Reproducibility: Yes

Additional Feedback:


Review 3

Summary and Contributions: The paper proposes a data poisoning attack to measuring a lower bound of the epsilon of differential privacy. The attack is clipping-aware and thus would give tighter lower bound than usual data poisoning attack. The paper then presents an experimental evaluation of the attack on common datasets, showing a roughly 10x improvement than previous attacks.

Strengths: The tightness of the DP epsilon in DPSGD has long been a question for practitioners and theory researchers. It is very nice to see an attack that provides a better lower bound.

Weaknesses: I don't see obvious weaknesses. Here are some general questions about the paper: - Is it computationally feasible to try multiclass problems than only binary? If so, would we expect to get better or worse lower bound? - Since DP is a data-independent definition, do you have any comment on what kind of datasets might be easier/harder to attack? - Regarding the effect that decreasing clipping norm would decrease epsilon_LB for random initialization, can that possibly be just an effect of the design of the attack?

Correctness: The claims and method seems correct to me.

Clarity: Yes. The paper is well written.

Relation to Prior Work: Yes.

Reproducibility: Yes

Additional Feedback:

[Author Response · NeurIPS 2020]

We would like to thank the reviewers for their effort in reviewing and providing feedback during these hectic times.

**Reviewer 1**

**Q**: Missing Carlini et al. USENIX? **A**: We omitted this accidentally, but will definitely reference this in our revised
version. Carlini et al. demonstrate privacy risks on models trained with standard SGD. Their attacks do not hold
even with very weak differential privacy guarantees. Our work, by contrast, designs attacks that are effective against
DP-SGD, and also shows how to use these attacks to better understand privacy guarantees offered by DP-SGD.

**Q**: Understanding of variance overfits to logistic regression **A**: Our attack works on multiple models, not only logistic
regression. In fact, we also evaluate our attack using two-layer neural networks, and the performance is similar. See
Figure 2 (d), (e), (f), and Tables 1 and 3.

**Q**: What does it mean for $y_p$ to be the smallest probability class on $x_p$? **A**: The class which the model predicts with
the smallest probability. If the model's class predictions are [0.4, 0.1, 0.5] (for classes 0, 1, and 2, respectively) then
$y_p$ would be class 1. We can update the paper to clarify this.

**Q**: Why does $\varepsilon = \infty$ use clipping? **A**: For consistency, we use clipping for all our experiments. Even with clipping,
setting the noise $\sigma = 0$ would still not give $\varepsilon$-DP for any finite $\varepsilon$. Additionally, Secret Sharer (Carlini et al., USENIX)
demonstrates the privacy risks of training without clipping/noise, and shows that gradient clipping already prevents
their identified privacy leakage. Our results establishing a privacy risk when gradient clipping is used are new.

**Q**: Fit to security venue? **A**: It's true that many papers studying privacy attacks appear in security venues, but we
believe our work is also solidly in scope for NeurIPS because it addresses questions that practitioners of machine
learning have to answer: which learning algorithm should they choose and how to set its privacy level in order to
achieve best trade-offs between privacy and accuracy.

**Q**: Why use multiple poisoning points and not the least $k$ singular vectors? **A**: Using different singular vectors as
poisoning would impact the model in uncorrelated ways. Using $k$ copies of the same data point causes the gradients
of each attack point to be correlated, making the poisoned and unpoisoned models more distinguishable.

**Q**: Why is membership inference not monotone? **A**: Despite averaging results over 10 trials for membership inference,
we still observe some variance. Note that the inferred lower bounds are very small for membership inference and we
are plotting the results in log scale, which might amplify the variance.

**Reviewer 2**

**Q**: Focus on SGD **A**: SGD (and other stochastic first-order methods) are among the most widely used methods for
private ML, and one of the few with high quality public implementations, so we felt this was the most important single
algorithm to study. However, in the supplementary material we also show that a suitable variant of our attack is highly
effective against the output perturbation algorithm for private linear regression (Chaudhuri et al. JMLR '11). More
generally, our underlying approach is applicable for any class of models for which poisoning attacks are known, which
currently includes linear models (logistic regression, SVM) and non-linear models (decision trees, neural networks).

**Q**: Scale to bigger datasets/more complicated architectures? **A**: Our poisoning attack works for any fixed-
dimensionality input. Our approach does still require training multiple models, which can be expensive as archi-
tectures get larger. Tuning the confidence parameter and number of poisoning points make it possible to observe
privacy leakage with fewer training runs. The architectures we consider are state-of-the-art for DP training.

**Q**: Work on transformers/RNNs? **A**: RNN-specific techniques are required for training DP-RNNs, and so our attacks
need to be modified accordingly. To our knowledge, DP transformers have not been considered in the literature.

**Reviewer 3**

**Q**: Multiclass problems? **A**: Our methods handle multi-class problems. In fact, we tested our techniques on all 100
classes of Purchase-100, and achieve a large lower bound.

**Q**: Worst case dataset? **A**: For output perturbation, our theoretical analysis suggests that datasets with small variance
directions will allow for better lower bounds than more spherically distributed datasets. Finding the worst-case dataset
is an interesting problem for future research in this space.

**Q**: Could impact of clipping norm be attack specific? **A**: We know that with clipping norm 0, the algorithm satisfies
0-DP, and with clipping norm $\infty$, the algorithm is wildly non-private (Carlini et al., USENIX). So it is reasonable to
expect that the clipping norm affects the true privacy level in less extreme parameter regimes, although we do not have
a rigorous justification for it. We note that our work is the first to identify the clipping norm as a relevant parameter,
and we expect this issue will attract future study regardless of what the answer ultimately proves to be.

[Meta-Review · NeurIPS 2020]

All three reviewers support acceptance of this paper. They agree that providing lower bounds on the privacy guarantees of DP-SGD is an important problem and that the paper makes significant headway on this problem. I therefore recommend accept. It would be worthwhile to incorporate changes to the camera ready version of the paper clarifying some of Reviewer 1's questions.